

# Morphological convergence in 'river dolphin' skulls

Charlotte E. Page[1] and Natalie Cooper[2]

[1] Department of Life Sciences, Imperial College London, London, United Kingdom
[2] Department of Life Sciences, Natural History Museum, London, United Kingdom

## ABSTRACT

Convergent evolution can provide insights into the predictability of, and constraints on, the evolution of biodiversity. One striking example of convergence is seen in the 'river dolphins'. The four dolphin genera that make up the 'river dolphins' (*Inia geoffrensis, Pontoporia blainvillei, Platanista gangetica* and *Lipotes vexillifer*) do not represent a single monophyletic group, despite being very similar in morphology. This has led many to using the 'river dolphins' as an example of convergent evolution. We investigate whether the skulls of the four 'river dolphin' genera are convergent when compared to other toothed dolphin taxa in addition to identifying convergent cranial and mandibular features. We use geometric morphometrics to uncover shape variation in the skulls of the 'river dolphins' and then apply a number of phylogenetic techniques to test for convergence. We find significant convergence in the skull morphology of the 'river dolphins'. The four genera seem to have evolved similar skull shapes, leading to a convergent morphotype characterised by elongation of skull features. The cause of this morphological convergence remains unclear. However, the features we uncover as convergent, in particular elongation of the rostrum, support hypotheses of shared feeding mode or diet and thus provide the foundation for future work into convergence within the Odontoceti.

## INTRODUCTION

Convergent evolution, or convergence, is the independent evolution of similar phenotypes in different lineages (*Losos, 2011*), producing taxa that are more similar than expected given their phylogenetic relatedness (*Conway Morris, 2008*). Convergence is widespread (*Conway Morris, 2003*; *McGhee, 2011*) and continues to be a central concept in evolutionary biology through both its role in describing evolutionary patterns and in providing strong evidence for natural selection (*Donley et al., 2004*; *Foote et al., 2015*; *Losos, 2011*; *Muschick, Indermaur & Salzburger, 2012*). Although convergent evolution has been studied since *Darwin (1859)*, there has been a recent resurgence of interest in the field, partly fueled by the ongoing debate on its role in limiting biodiversity (*Mahler et al., 2013*; *Speed & Arbuckle, 2016*). If the evolutionary forces that cause convergence are common, then phenotypes of organisms may be predictable, ultimately constraining the diversity of living species (*Conway Morris, 2003*; *Conway Morris, 2008*; *Losos, 2011*).

Corresponding author
Charlotte E. Page,
c.page17@imperial.ac.uk,
charlotte.eve.page@gmail.com

Despite recent developments in methods for quantifying of convergence (e.g., *Arbuckle, Bennett & Speed, 2014*; *Ingram & Mahler, 2013*; *Speed & Arbuckle, 2016*; *Stayton, 2015a*), convergence is rarely rigorously quantified beyond some classical examples; such as *Anolis* lizards (e.g., *Mahler et al., 2013*) and cichlid fishes (e.g., *Muschick, Indermaur & Salzburger, 2012*). Quantitative analysis of convergence in more taxa will help us to gain further understanding of the concept and the mechanisms that underlie it. It will also allow exploration of whether qualitative human classifications of convergence using external morphology alone are quantitatively justified.

One iconic example of convergent evolution is in the 'river dolphins', a group of distantly-related cetaceans that secondarily entered river and estuarine systems from the ocean, and evolved similar morphological characteristics (*Hamilton et al., 2001*). The river dolphins consist of four recent genera: the Amazon river dolphin (*Inia geoffrensis*), the La Plata dolphin (*Pontoporia blainvillei*), the Yangtze river dolphin (*Lipotes vexillifer*), which is thought to be extinct (*Turvey et al., 2007*) and the Ganges river dolphin (*Platanista gangetica*), which contains two subspecies, *Platanista gangetica spp. gangetica* and *Platanista gangetica spp. minor* (*Kasuya & Aminul Haque, 1972*) (Fig. 1A). *Platanista*, *Lipotes* and *Inia* are found exclusively in riverine systems, whereas *Pontoporia* is a coastal and estuarine species. Note that there are also other odontocete genera not included in the 'river dolphin' grouping that are found in riverine systems, such as the Irawaddy dolphin (*Orcaella brevirostris*) and the Tucuxi dolphin (*Sotalia fluviatilis*) (*Borobia et al., 1991*; *Stacey & Leatherwood, 1997*) but we do not consider these species here.

Based on their similar external morphology and shared riverine and estuarine habitats, taxonomists originally placed *Platanista*, *Lipotes*, *Inia* and *Pontoporia* into a single higher grouping, the Platanistoidea (*Kasuya & Aminul Haque, 1972*; *Simpson, 1945*). Molecular techniques have since clarified that the four taxa do not form a single monophyletic group (*Geisler et al., 2011*; *Hamilton et al., 2001*; *McGowen, Spaulding & Gatesy, 2009*; *Steeman et al., 2009*). Analysis consistently group *Inia* and *Pontoporia* as sister taxa (*Cassens et al., 2000*; *Geisler et al., 2011*; *Hamilton et al., 2001*; *Messenger & McGuire, 1998*; *Steeman et al., 2009*) with *Lipotes* as a sister-group (*Geisler et al., 2011*; *McGowen, Spaulding & Gatesy, 2009*; *Steeman et al., 2009*). The most supported placement of *Platanista* is as a sister-group to all other odontoceti, excluding Physeteridae (sperm whales) and Kogiidae (pygmy sperm whales) (*Hamilton et al., 2001*; *McGowen, Spaulding & Gatesy, 2009*; *Steeman et al., 2009*) (Fig. 1A). Convergent features of the river dolphins include a longirostral skull (i.e., a long narrow rostrum and mandible), an elongated and fused mandibular symphysis, relatively more teeth than in other dolphin lineages (up to 250 teeth in *Pontoporia*, compared to 100 in *Tursiops* (*Werth, 2006*)), an extended alveolar tooth row, long zygomatic process, nasal bones that lie at the same level as the squamosal processes (Fig. 1B), and a flexible neck due to unfused cervical vertebrae (*Geisler & Sanders, 2003*). They also share a number of soft anatomical features such as broad forelimb flippers and reduced eyes (*Cassens et al., 2000*) (Fig. 1B).

Several studies have identified a river dolphin morphotype, based on either discrete measurements (*Werth, 2006*) or through the use of geometric morphometrics (*Barroso, Cranford & Berta, 2012*; *McCurry et al., 2017a*). Convergence among the river dolphins has
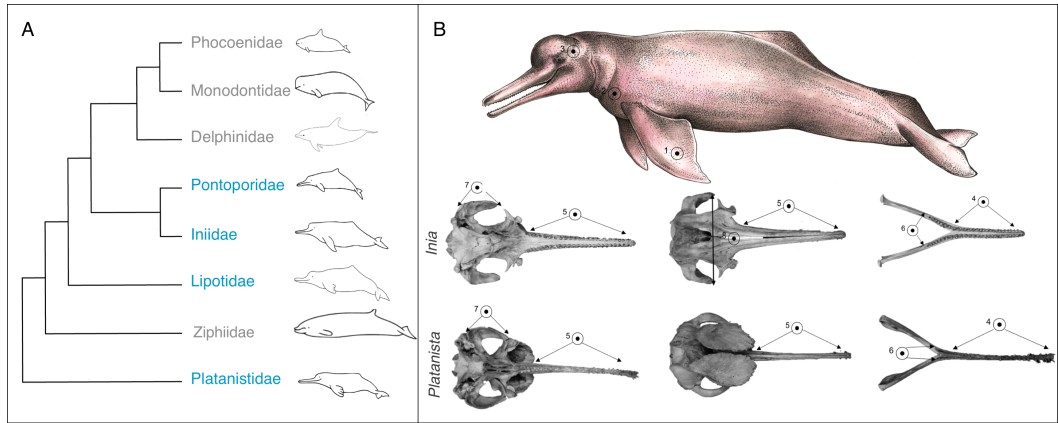

**Figure 1** **Phylogeny and convergent features of the 'river dolphins'.** (A) Molecular phylogeny of Odontoceti (adapted from *Steeman et al., 2009*). This topology places *Lipotes* as a sister-group to Iniidae + Pontoporidae. (B) Convergent features (indicated by numbered circles) of two 'river dolphin' skulls, *Inia geoffrensis* (top, NHMUK_1169.b), and *Platanista gangetica* (bottom, NHMUK_1884.3.29.1). Soft anatomical features are shown on a photograph of *Inia geoffrensis* (top row). The views of the skull are: dorsal view of the cranium, ventral view of the cranium and dorsal view of the mandible. Features are as follows: (1) broad forelimb flippers, (2) reduced eyes, (3) flexible neck, (4) elongated and fused mandibular symphysis, (5) elongated rostrum, (6) extended alveolar tooth row, (7) long zygomatic arches and (8) nasals in line with zygomatic processes. Feature 8 is not visible on *Platanista* because the maxillary crests project anteriorly over the cranium and hide the nasals. Skulls are not to scale. Cetacean outlines by Chris huh licensed under CC BY-SA 3.0 (https://commons.wikimedia.org/wiki/File:Cetaceans.svg). River dolphin illustration by Francesca Page licensed under a CC attribution 4.0 international license.

also indirectly been tested for through the quantification of convergence between the four river dolphin genera and gharials, a riverine species of crocodile (*McCurry et al., 2017a*).

Here we present an investigation into the morphological variation present in the skulls of river dolphins. We use geometric morphometric (GMM) techniques (*Rohlf & Marcus, 1993*) to compare skull morphology of the river dolphins to a wide sample of extant Odontoceti (toothed whale and dolphin) genera. We then apply multiple tests for convergence and provide a detailed analysis of cranial and mandibular features which are convergent among the genera. Our results reveal that the river dolphins show significant convergence in the shape of their crania and mandibles when compared to other odontocete species. We uncover a number of convergent features that are characterised by elongation of the skull.

## MATERIALS AND METHODS

### Data collection

One of us (CP) collected data from the Natural History Museum, London. We photographed all available, complete river dolphin specimens (four species, crania: 12 specimens, mandibles: 10 specimens; Table S1). We chose other taxa to sample using the phylogeny of *Steeman et al. (2009)*. This places *Lipotes* as a sister group to *Inia* and *Pontoporia*. Monodontidae (belugas and narwhals), Phocoenidae (porpoises) and Delphinidae (dolphins) together form the sister-group to *Inia* and *Pontoporia*, plus *Lipotes*, whilst *Platanista* forms a sister-group to these odontocetes plus Ziphiidae. We sampled

species ($n = 24$, Table S1, Fig. S1) across all groups except Physeteridae (sperm whales) and Kogiidae (pygmy sperm whales), which are more distantly-related, and either too large to sample using our protocol (Physeteridae) or so rare that NHM has no samples (Kogiidae). Sexual dimorphism varies among odontocete genera, being present in some species as differences in size (*Amaral et al., 2009*; *Higa, Hingst-Zaher & Vivo, 2002*). The effects of sex on morphology varies in different species; for example no sex differences are seen in *Pontoporia* (*Higa, Hingst-Zaher & Vivo, 2002*), but they are observed in some species of *Mesoplodon* (*Besharse, 1971*). We therefore chose males and females from different collection locations where possible to provide a representative sample of each species. Juveniles were not included because skull characters typically change during growth and development (*Perrin & Heyning, 1993*). Juvenile specimens were marked in the collections on their specimen labels, and could also be identified by the incomplete fusion of their skull bones. See Table S1 for full details of the specimens used and their accession numbers. All data are available from the Natural History Museum's Data Portal at http://dx.doi.org/10.5519/0082274 (*Page & Cooper, 2017a*).

We adjusted a protocol described by Báez-Molgado and colleagues (*2013*) and photographed specimens using a Canon EOS 550D fitted with a EFS 18–55 mm lens. To account for variations in lighting, we used a white card to set the custom white-balance function on the camera at the start of each session. We included a 25 cm scale bar and specimen accession number in every photograph. We placed the specimens directly under the camera lens and used foam board to make sure the specimen was level. A problem with imaging is the phenomenon of parallax, which occurs when a camera lens is placed too close to a specimen, thereby producing a slightly warped or distorted image. However, the error produced by this phenomenon is constant among samples when the same lens orientation and positioning is used (*Mullin & Taylor, 2002*). For this reason, the same photographing setup was replicated at every photographing session.

For each specimen we photographed (1) ventral view of the cranium (77 specimens; 24 species) and (2) dorsal view of the mandible (67 specimens; 23 species). The numbers of specimens in the two analyses varied because some specimens had damaged mandibles. After photographing in raw file format, we converted the photographs to grey-scale to help with structure identification, and exported them as TIFFs. We then converted these files into import TPS files using the tpsUtil (*Rohlf, 2010*) 'build TPS files from images' function.

## 2D Geometric morphometric analyses (GMM)

We used 2D GMM to capture the shape of the dolphin skulls (*Mitteroecker & Gunz, 2009*). We used a combination of landmarks and semi-landmarks. Where possible, we used landmarks that had been previously used in the cetacean literature, but we primarily chose landmarks based on the objective of this study, i.e., placing emphasis on putatively convergent features of the river dolphin skulls outlined in Fig. 1B (*Geisler et al., 2011*; *Geisler & Sanders, 2003*). We focused on the ventral view of the cranium because this view allowed us to choose homologous landmarks across all species, whilst also focusing on putatively convergent features (Figs. 1 and 2). To remove errors associated with using a 2D image, we chose landmarks that were in the same plane. One of us (CP) digitised all

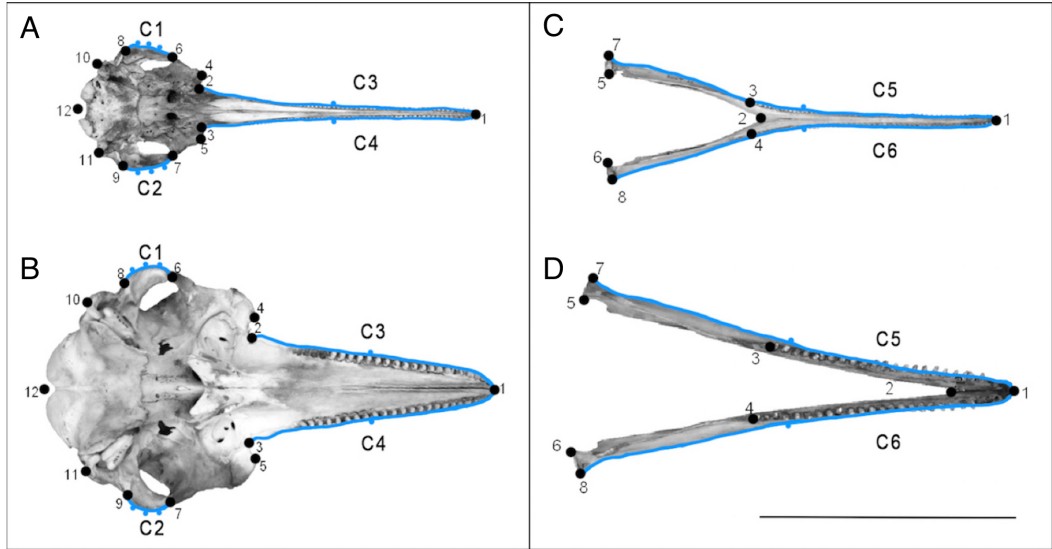

**Figure 2 Landmarks used on specimens.** Landmarks (numbered points) and curves with semi-landmarks (lettered blue outlines with points) for the ventral view of the cranium (A) and the dorsal view of the mandible (B). Specimen A and C is *Pontoporia blainvillei* (NHMUK_1925.11.21), specimen B and D is *Tursiops truncatus* (NHMUK_1960.5.11.10). The line represents the 25 cm scale bar. Descriptions of the landmarks can be found in Tables S2 and S3.

landmarks shown in Fig. 2 using tpsDig (*Rohlf, 2006*), on separate data files for each view. We set the scale on each image individually. Detailed descriptions of the landmarks can be found in Tables S2 and S3. We digitised 12 landmarks onto the images depicting the ventral view of the cranium (Figs. 2A and 2B, Tables S2). These were based on GMM studies of odontocete and river dolphin genera (*Amaral et al., 2009*; *Higa, Hingst-Zaher & Vivo, 2002*). We digitised eight landmarks onto the images depicting the dorsal view of the mandible (Figs. 2C and 2D, Table S3). These were adapted from GMM studies that consider the dorsal mandibular view of Odontoceti (although these used 3D images; Table S4) (*Barroso, Cranford & Berta, 2012*; *McCurry et al., 2017a*; *McCurry et al., 2017b*).

We also drew curves on each specimen before resampling them with a specified (Figs. 2A–2D, Tables S2–S5) number of equally spaced semi-landmarks. The semi-landmark approach can introduce error into GMM analyses through oversampling curves, because simpler structures, such as the rostrum, will require fewer semi-landmarks to accurately represent their shape (*MacLeod, 2012*), compared to more complex structures. To overcome this, we followed a re-sampling method described by *MacLeod (2012)* to determine the minimum number of semi-landmarks needed to measure an outline to at least 95% accuracy of the true length of the curve. We resampled cranial curves C1 and C2 with five points, and curves C3 and C4 with three points. We resampled mandibular curves C5 and C6 with three points. See Fig. 2 and Supplemental Information 2 for more details.

We saved the landmark coordinates as a TPS file, and downloaded them into R package version 3.0.4 to carry out all further analyses (*R Core Development Team, 2017*). We carried out separate analyses on both the cranial and mandibular datasets at all stages. We used

the 'gpagen' function in geomorph (*Adams et al., 2017*) to run a Generalised Procrustes alignment (GPA). This converts the digitised, raw landmark configurations into shape variables by removing non-shape (i.e., scale, rotation and size) variation (*Rohlf & Marcus, 1993*). Because we are interested in convergence among species, we then calculated the mean GPA coordinates for each odontocete species, and used these for all further analysis. We also repeated the analyses using the specimen-level GPA coordinates and report these results in Supplemental Information 3. We used the phylogeny of *Steeman et al. (2009)* in all phylogenetic analyses (Fig. 1A).

### Error checking

Error can be introduced at two main points of a GMM analysis: (1) photographing, and (2) digitising (*Zelditch, Swiderski & Sheets, 2012*). Error present in GMM datasets can affect later analyses by inflating the amount of variance among samples, and obscuring biological signal (*Fruciano, 2016*). To assess the measurement error in our data, we followed a method outlined by *Zelditch, Swiderski & Sheets (2012)* on replicate attempts to calculate the intraclass correlation coefficient (*Fisher, 1925*), often termed 'repeatability' (*Fruciano, 2016*). See Supplemental Information 2 for more detail.

### Exploring shape variation in dolphin skulls

To uncover the shape variation present in the skulls of the river dolphins, we conducted a principal component analysis (PCA) on the mean species GPA coordinates for both the cranium and mandible datasets using the geomorph function, 'plotTangentSpace' (*Adams et al., 2017*). We then visualized patterns of cranial and mandibular shape evolution in phylomorphospace by plotting PC axes 1–3 against each other and projected phylogeny onto the morphological trait space (*Sidlauskas, 2008*) using the function 'plotGMPhyloMorphoSpace' in geomorph (*Adams et al., 2017*). In these plots, each data point represents a shape, and species close to each other in the morphospace are more similar in shape. Shape changes along the PC axes correspond to landmark eigenvectors, which we visualised using wire frame deformation grids, using the function 'plotRefToTarget' in geomorph (*Adams et al., 2017*). It is important to note that we did not use a phylogenetic PCA (pPCA) to plot the phylomorphospaces. This is because pPCA scores are correlated across axes, unlike PC axes (*Polly et al., 2013*), so cannot be used for further analysis.

### Testing for convergence

We first estimated phylogenetic signal in the mean species cranial and mandibular GPA landmarks using $K_{mult}$ (*Adams, 2014*), the multivariate version of Blomberg's $K$ (*Blomberg, Garland & Ives, 2003*). Phylogenetic signal was present in both our datasets (cranium: $K_{mult} = 1.14$, $p = 0.001$; mandible: $K_{mult} = 1.17$, $p = 0.001$), so we used analytical methods that account for phylogeny in our tests for convergence.

To determine whether river dolphins have significantly different skull shapes compared to other odontocetes, we performed a Procrustes ANOVA while accounting for phylogenetic relatedness, on the mean species GPA coordinates for both cranial and mandibular datasets using the function 'procD.pgls' in geomorph (*Adams et al., 2017*) with 1,000 iterations.

This analysis tells us about the overall shape variation present. Therefore, to identify which specific shape axes are important, we performed multiple phylogenetic ANOVAs for each dataset on the PC axes which accounted for >95% of the variation present in both the cranium and mandible (PC1–PC4 for crania and PC1–PC3 for mandibles), using the 'aov.phylo' function in the geiger R package (*Harmon et al., 2007*).

Next we constructed phenograms for the crania and mandible datasets and compared these to the phylogeny. We used Ward's hierarchical clustering agglomerative method (*Ward jr, 1963*) on a distance matrix generated using the PCs accounting for >95% of the variance in shape (PC1–PC4 for crania and PC1–PC3 for mandibles) to build the phenetic trees. Ward's method considers all possible species pairs of clusters, and merges those that result in the minimum increase in the error sum of squares (*Ward jr, 1963*). Species that cluster together will therefore have the most similar morphology.

Finally, we quantified the amount of convergence in both cranium and mandible datasets using a distance-based approach (*Stayton, 2015a*). This method is based on the idea that convergence occurs when two taxa evolve to be more similar than their ancestors were to one another (*Losos, 2011*; *Stayton, 2015a*) and produces an index of convergence (C1). We calculated C1 for both the cranium and mandible datasets using the PCs accounting for >95% of the variance in shape (PC1–PC4 for crania and PC1–PC3 for mandibles) whilst also meeting the statistical requirement of fewer shape variables than putatively convergent taxa ($n = 4$) using the R package convevol (*Stayton, 2015b*). The function 'convrat' infers ancestral states using weighted means of extant species data and also scales C1 to permit comparisons among different taxa (*Stayton, 2015b*). We tested the significance of each C1 calculated using the function 'convratsig' (*Stayton, 2015b*). R code for all analyses is available from http://dx.doi.org/10.5281/zenodo.846278 (*Page & Cooper, 2017b*).

## RESULTS

### Error checking

The level of error in our results was negligible. Repeatability was 91.3% in the cranial dataset and 93.1% in the mandibular dataset (Supplemental Information 2).

### Exploring shape variation in dolphin skulls

River dolphins cluster together in morphospace, but there was considerable shape variation across the odontocetes (Fig. 3). More than 95% of the variation in shape is explained by the first four PC axes for odontocete crania, and the first three PC axes for odontocete mandibles. The variance explained by each major PC (>95% variation) and the loadings of the landmarks on each PC axis (eigenvectors) are in the Tables S6–S8.

Shape changes associated with these PC axes are depicted by wireframe deformation grids at the minimum and maximum extent of each PC axis in Fig. 4. Cranial PC1 describes variation in the shape of the rostrum and the cranium (Fig. 4A). This represents the relative changes in the positions of the tip of the rostrum, and the rest of the cranium (Fig. 4A). Shape changes associated with the maximum extent of PC1 are an elongation and narrowing of the rostrum, and narrowing of the cranium. PC2 describes variation in the shape of the

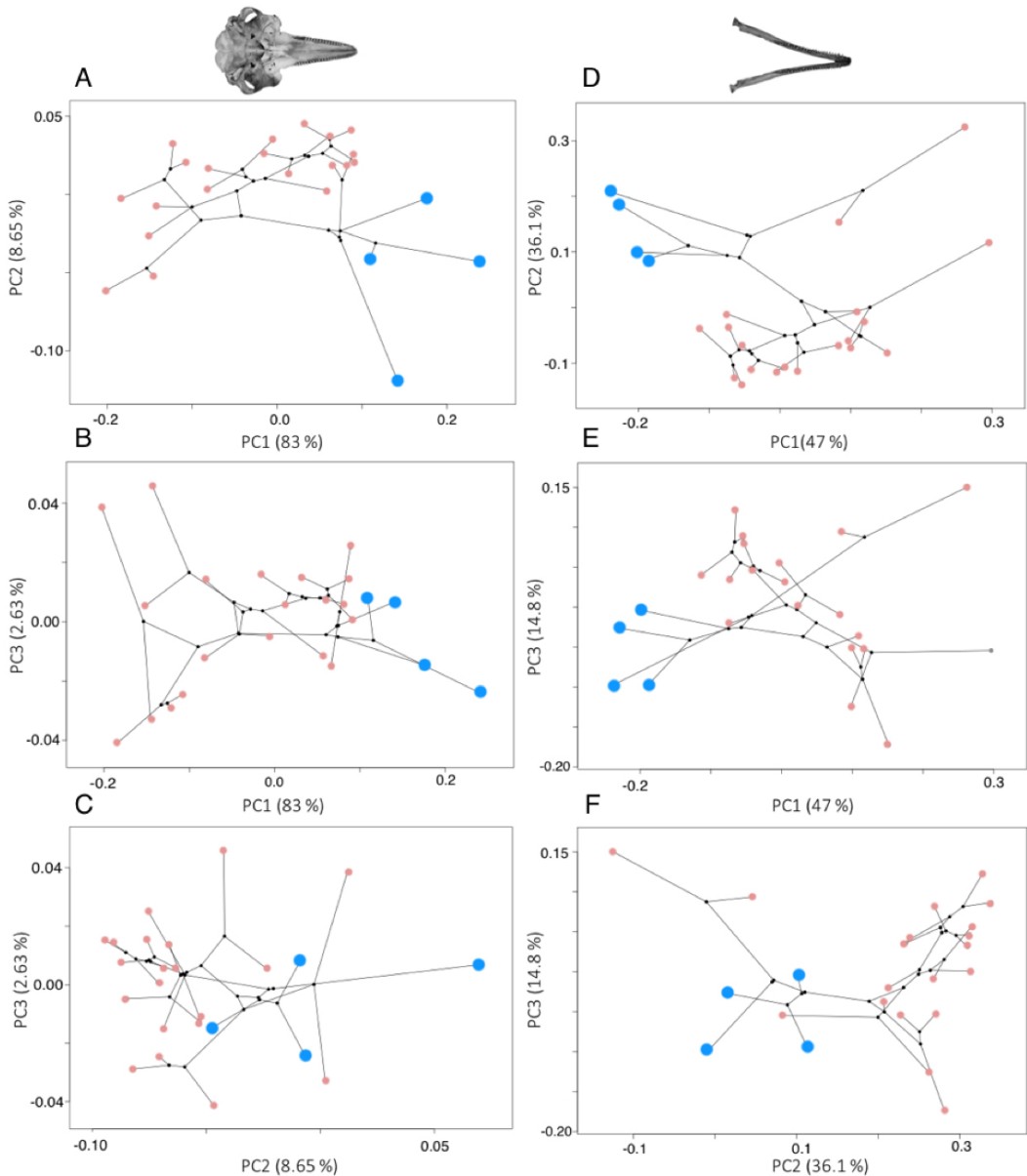

**Figure 3 Phylomorphospace plots for cranium (A–C) and mandible (D–F) morphology.** In both plots the river dolphin species are represented by blue points ($n = 4$), pink points represent other odontocetes (cranium: $n = 20$, mandible: $n = 19$), and black points represent internal ancestral nodes. Each point represents the average cranium or mandible shape of an individual species and lines represent the phylogenetic relationships.

rostrum and zygomatic arches (Fig. 4B). PC3 and PC4 describe variation in the shape of the lacrimojugal bones, in addition to small changes in the shape of the rostrum (Fig. 4C).

Mandibular PC1 describes variation in the elongation of the mandible. Shape changes associated with the minimum extent of PC1 are an elongation of the rostrum, symphysis and alveolar tooth row (Fig. 4D). PC2 describes decreases in the length of the symphysis

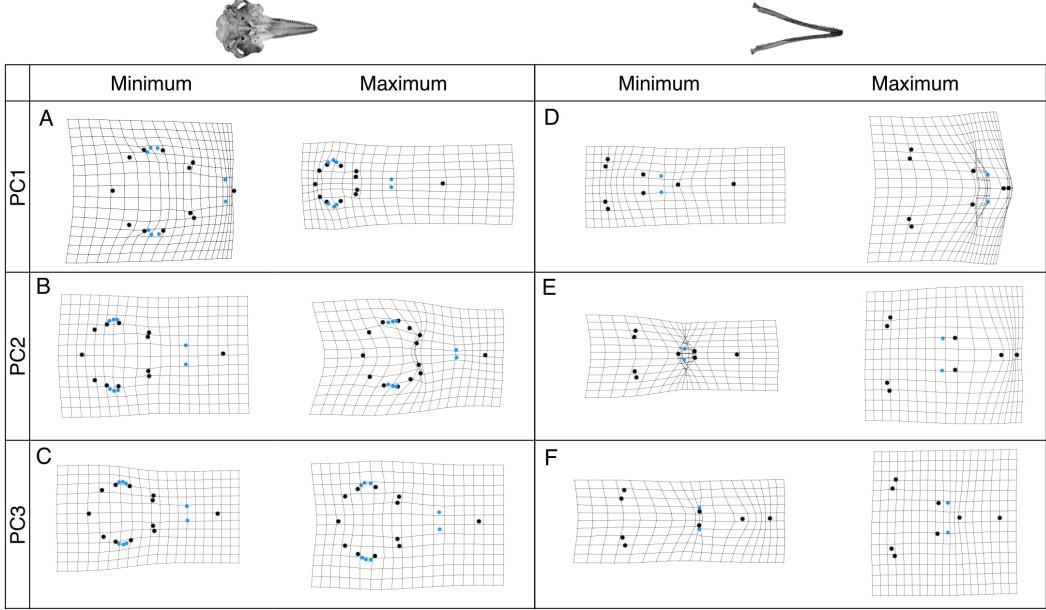

**Figure 4** **Wireframe deformation grids showing shape changes.** Grids represent the shape changes associated with the minimum and maximum extents of PC1, PC2 and PC3 for crania (A–C) and mandibles (D–F) are shown. Black points represent landmarks and blue points represent semi-landmarks digitised onto the cranium and mandible.

and increases in the alveolar tooth row length (Fig. 4E) PC3 describes variation in the length of the symphysis and slight changes in jaw flare (Fig. 4F).

## Testing for convergence

Overall, river dolphins have significantly different skull shapes compared to other odontocetes for both the cranium (phylogenetic Procrustes ANOVA: $F_{1,22} = 3.96$, $p < 0.001$) and mandible (phylogenetic Procrustes ANOVA: $F_{1,21} = 3.89$, $p < 0.001$) datasets. When considering individual PC axes, the four river dolphin genera occupy significantly different positions on mandibular PC1 (phylogenetic ANOVA: $F_{1,22} = 19.268$, $p < 0.05$) compared with other odontocetes, but there are no significant differences for the other PC axes in either the crania or mandible datasets (Table S9). The specimen-level, rather species-level, results show a similar pattern (Table S10).

Phenetic trees based on cranium (Fig. 5) and mandible (Fig. 6) major PCs cluster the river dolphins together. Within the river dolphins, the analysis pairs *Lipotes* with *Inia*, then *Platanista* and finally *Pontoporia*, based on skull morphology; and pairs *Lipotes* with *Inia*, and *Pontoporia* with *Platanista*, based on mandible morphology.

C1 values (*Stayton, 2015b*) indicate that the river dolphins evolved to be more similar to each other than would be expected under a null model of Brownian motion evolution (crania: C1 = 0.521, $p < 0.001$; mandibles C1 = 0.622, $p < 0.001$).

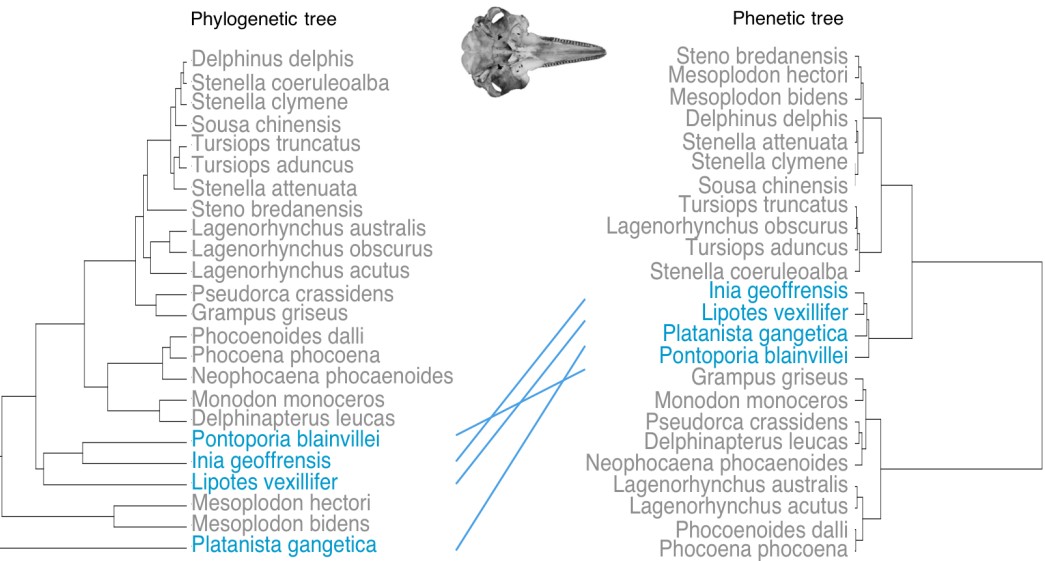

**Figure 5** **Tanglegram showing comparison in the position of the river dolphins on a phylogenetic tree and a phenetic tree based on cranial morphology.** Lines between trees link the same species and crossing lines indicate a lack of similarity in the two trees (e.g., where phenotype is more similar than implied by phylogeny, indicative of convergence). River dolphins are highlighted in blue.

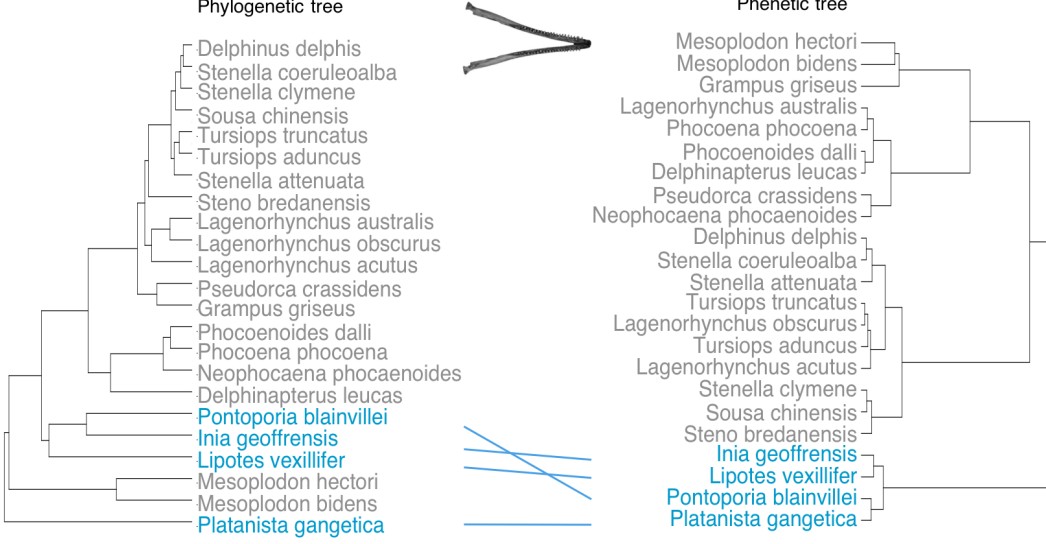

**Figure 6** **Tanglegram showing comparison in the position of the river dolphins on a phylogenetic tree and a phenetic tree based on mandibular morphology.** Lines between trees link the same species and crossing lines indicate a lack of similarity in the two trees (e.g., where phenotype is more similar than implied by phylogeny, indicative of convergence). River dolphins are highlighted in blue.

## DISCUSSION

Despite disparate phylogenetic histories, we find that the river dolphins exhibit similar variation in cranial and mandibular morphologies. The four genera seem to have evolved similar skull shapes (i.e., towards higher scores of PC1 for the cranium and towards lower scores of PC1 for the mandible). Collectively, morphospace positions of the river dolphins define a morphotype characterised by the elongation of skull features (rostrum, zygomatic arches, mandibular symphysis and alveolar tooth row) and narrowing of the brain case. Our results suggest that this morphotype is significantly convergent. We find that the river dolphins cluster differently based on morphology than they do on a phylogeny based on molecular data (*Steeman et al., 2009*). In particular, the position of *Platanista* changes so that it is convergent to *Inia, Lipotes* and *Pontoporia. Stayton's (2015a)* distance-based measure of convergence on both datasets, also suggests that the river dolphins are morphologically more similar to each other than their ancestors were.

The river dolphin morphotype uncovered is consistent with discrete characteristics that have been highlighted as convergent amongst the river dolphins (*Geisler & Sanders, 2003*). In particular, elongation of morphological features, a character named 'longirostral' (*Geisler & Sanders, 2003*) dominates shape changes associated with both the crania and mandibles. This shape change is also one that has been found by other studies on odontocete skulls (*Barroso, Cranford & Berta, 2012*; *McCurry et al., 2017a*; *Werth, 2006*). The river dolphins represent a polyphyletic group (*Geisler et al., 2011*; *Hamilton et al., 2001*; *McGowen, Spaulding & Gatesy, 2009*; *Steeman et al., 2009*) indicating that this morphotype has evolved more than once within the odontocetes. There are several other possible evolutionary explanations for the shared morphological characteristics uncovered in this study: they could be plesiomorphic (an ancestral trait shared by two or more taxa), they could be due to reversals, or some combination of these. Several authors have suggested that the narrow and elongate rostrum shared by *Platanista* and the other river dolphin genera, is in fact plesiomorphic (*Geisler & Sanders, 2003*; *Messenger & McGuire, 1998*). If shared characters were plesiomorphic, we may expect to see the sister species *Pontoporia* and *Inia* cluster together based on morphology, but this is not what we find. The placing of *Platanista* can have an impact on whether shared river dolphin characters are constructed as plesiomorphic (*Geisler et al., 2011*). However, if we accept the position of *Platanista* as a sister-group to all other odontoceti, excluding Physeteridae and Kogiidae (*Steeman et al., 2009*), it is more parsimonious to disregard plesiomorphy.

The cause of the convergence demonstrated here is still debated. Convergent evolution among other taxa has largely been attributed to adaptation to a similar niche (*Losos, 2011*; *Muschick, Indermaur & Salzburger, 2012*). Among the river dolphins, this includes utilization of riverine systems, mode of feeding and prey size (*Cassens et al., 2000*; *Geisler et al., 2011*; *Hamilton et al., 2001*; *Kelley & Motani, 2015*; *Werth, 2006*). Convergent evolution can occur for other reasons that are unrelated to adaptive evolution in similar environments (*Losos, 2011*; *Stayton, 2008*). However, key features of the river dolphin skull morphotype described by this study could be related to prey capture, and therefore adaptive evolution, i.e., elongate rostrum, mandibular

symphysis and zygomatic arches (*Cassens et al., 2000*). The river dolphins have been classified as raptorial feeders (*Werth, 2006*), and have a diet consisting of small and agile riverine fish (*Kelley & Motani, 2015*). Elongation of the skull significantly correlates with prey type (*McCurry et al., 2017a*; *McCurry et al., 2017b*; *Werth, 2006*) and has also been hypothesised as morphologically optimal for raptorial feeding through reduction in drag (*McHenry et al., 2006*). Similarity in diet has been shown to underlie the convergence seen amongst the river dolphins and gharials (*McCurry et al., 2017a*).

There were several limitations in our approach. Availability of skull material for some species was limited, with some species having only one specimen available i.e., for the now extinct species, *Lipotes* (*Turvey et al., 2007*). This led to our sample sizes being limited in some cases. The results presented here are also restricted to only cranial and mandibular skeletal morphology. Further work could look to corroborate this study through the analysis of other skeletal and soft anatomy traits. An analysis of 2D images to measure morphological variation in a 3D structure is a pitfall and inevitably there will be a loss of information (*Cardini, 2014*). However, the benefits compared to linear measurements in understanding river dolphin skull shape is great. Collection of 2D data is also inexpensive and fast, often leading to large sample sizes. Despite this, repetition of this study with 3D data could provide further insights into the morphological variation present amongst the river dolphin genera. For example, analysing the 3D shape of structures which differ in river dolphin species compared to other odontocete genera, may reveal subtle disparities that are missed in a 2D analysis.

*Stayton*'s *(2015a)* distance-based measure is a recently proposed method and the interpretation of C1 values varies among authors. The interpretation used in this study is conservative, following that used by *McLaughlin & Stayton (2016)* and *Stayton (2015a)*. This metric also relies upon accurate ancestral state reconstructions (*Stayton, 2015a*), which are calculated using weighted means of species data. This means that the phenotypes of ancestors are restricted to fall between the sampled extant species values, when in reality ancestors may occur outside this phenotypic space.

## CONCLUSIONS

Here we have presented a quantitative investigation into convergence in the river dolphins. Our results corroborate those of other studies (*Barroso, Cranford & Berta, 2012*; *McCurry et al., 2017a*; *Werth, 2006*) and show that overall skull morphology of the river dolphins is significantly convergent, being more similar than expected given their phylogenetic relationships. We find that *Platanista* shares the 'river dolphin' morphotype with the other river dolphin genera. This morphotype is characterised by the elongation of the rostrum, mandibular symphysis and zygomatic arches. All of these features are related to the 'raptorial' mode of feeding adopted by the river dolphin genera (*McCurry et al., 2017a*; *McCurry et al., 2017b*; *McHenry et al., 2006*; *Werth, 2006*). The findings we present provide the foundation for future work into convergence within the Odontoceti, in addition to quantitatively justifying qualitative human classifications of the river dolphin grouping based on morphology alone.

## ACKNOWLEDGEMENTS

We thank Richard Sabin for assistance with accessing the cetacean collections at NHM, Ellen Coombs for help handling larger skulls and Dan Bell for use of his photos. We also thank Marcela Randau for helpful comments on earlier drafts of this work. Francesca Page also enriched our work with an illustration of *Inia geoffrensis*.

### Funding

The authors received no funding for this work.

### Competing Interests

The authors declare there are no competing interests.

### Author Contributions

- Charlotte E. Page conceived and designed the experiments, performed the experiments, analyzed the data, wrote the paper, prepared figures and/or tables, reviewed drafts of the paper.
- Natalie Cooper conceived and designed the experiments, performed the experiments, analyzed the data, contributed reagents/materials/analysis tools, wrote the paper, reviewed drafts of the paper.

### Data Availability

NHM Data Portal: http://dx.doi.org/10.5519/0082274.

Code (and data):

GitHub: https://github.com/NaturalHistoryMuseum/river-dolphin-convergence.
https://doi.org/10.5281/zenodo.846278.

### Supplemental Information

Supplemental information for this article can be found online at http://dx.doi.org/10.7717/peerj.4090#supplemental-information.

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
