# Peer review of "Morphological convergence in ‘river dolphin’ skulls"

_PeerJ, doi:10.7717/peerj.4090_

## Round 0.1 · original submission · Major Revisions

Both reviewers have provided extensive comments on the manuscript. I am in agreement with them that this is a generally well-written manuscript with carefully designed and executed methods. However, there are a number of issues with the framing of the study and the interpretation of results that need to be addressed, thus I have given a decision of major revisions.

Please can you attend to all the comments made by the reviewers. In addition, I have a couple of points to raise:

1. 'Morris 2003 & 2008' should be 'Conway Morris' - Conway is part of the surname, not a middle name.

2. Line 52. You state that the 4 river dolphins do not form a monophyletic group. However, 3 of them do. I think this should be noted here - the sentence as it currently stands exaggerates the difference between them.

3.Discussion - since you only used 2D GMM, it would be good to see a brief consideration of whether 3D data would have altered your result at all, or provided useful information.

4. I agree with reviewer 2 that figures 3 and 4 are not easily legible at the moment. Increasing the size of the dots, and using more contrasting colours in figure 3 would help.

I look forward to seeing a revised version of the manuscript.

Reviewer 1 ·

Basic reporting

Summary: Charlotte E Page and Natalie Cooper present a 2D geometric morphometric analysis of odontocete skulls with the aim of testing for convergence in river dolphins. The dataset contains 24 species with between 1 and 5 individuals of each species. The methods appear sound and thorough e.g. the checks of repeatability and multiple tests of convergence. I have a number of issues with the reporting that I think should be improved prior to publication.

Line edits:

Title: the sub title “a disparate grouping justified” generates a little confusion because the term “disparate” is often used to describe morphologically dissimilar species. Please consider rephrasing to a clearer title.

Line 6: I think the novelty of this study is overstated within the abstract. You state: “This has led many to using the ‘river dolphins’ as an example of convergent evolution. However, these morphological similarities have never been quantified.” McCurry et al. 2017a quantifies morphological similarities between a wide range of crocodilians and odontocetes, including each of the river dolphin species.
Please re-write the introduction as to clarify what the knowledge gap is, and how this study fills it.

Line 13: “The cause of this morphological convergence remains unclear, but our results support hypotheses of shared feeding mode or diet and thus provide the foundation for future work into convergence within the Odontoceti.”
Please state exactly how your results support hypotheses of the cause of convergence?

Line 24: Why is the “tree of life” written in Caps?

Line 42: “One iconic example of convergent evolution is in the ‘river dolphins’, a group of distantly related cetaceans that secondarily entered river systems from the ocean, evolving riverine lifestyles and similar morphological characteristics (Hamilton et al. 2001).”
The term “river dolphin” is more convoluted than this. Pontoporia is not a riverine species, inhabiting both estuaries and oceanic environments. Furthermore, some dolphins outside of the “river dolphin” grouping are found in riverine environments e.g. Sotalia fluviatilis. Please rewrite this section in a way that describes the complexity of habitat similarities/differences.

Line 62: “Here we present the first quantitative investigation of the morphological variation present in the skulls of river dolphins, and quantify whether the four river dolphin genera are convergent.”
This statement ignores past work, please fit Barroso et al. and McCurry et al. into the introduction and explain what you do differently to these studies.

Line 75: What do you mean by “equally closely related to river dolphins”, this needs more clarification. Why did you include the species that you did?

Line 76: “we photographed specimens across all Odontoceti” please state the number of species sampled here, otherwise it sounds like you sampled all species within the group. You could even include an odontocete tree showing which species were sampled and the sample size for each species?

Line 78: “Sexual dimorphism varies among odontocete genera, being present in some species as differences in size, but with no differences in shape between sexes recorded (Amaral et al. 2009; Higa et al. 2002). We therefore chose males and females from different collection locations where possible to provide a representative sample of each species.”
This is confusing, do you mean that some species have sexual dimorphism in size and other have differences in morphology? Or that only differences in size exist? Some odontocetes do show variation in cranial morphology dependant on sex e.g. Mesoplodon (Besharse 1971), monodon etc..
Please rewrite to clarify.
Besharse, J. C. (1971). Maturity and sexual dimorphism in the skull, mandible, and teeth of the beaked whale, Mesoplodon densirostris. Journal of Mammalogy, 52(2), 297-315.

Line 106: Please state here that it was 2D GMM that was used.

Line 109: Which “putatively convergent features”? Please provide more detail.

Line 133 Is there a full stop missing after the reference?

Line 225: “Mandibular PC1 describes variation in the elongation of the mandible resulting in an elongation of the rostrum, symphysis and alveolar tooth row (Figure 4D)”.
The word use of the word “resulting” here inappropriately implies cause/effect.

Line 253: The four genera seem to have experienced evolution in the same direction.
Consider rephrasing to “evolved similar skull shapes”.

Line 296: What are the limitations/benefits of 2D vs 3D GMM? Please consider adding this to this paragraph.

Line 299: “Skeletals s” is a typo.

Line 313: “Here we have presented the first quantitative investigation into convergence in the river dolphins.”
As stated earlier, this is not the first study to use GMM on a sample including river dolphins, or to identify convergence.

Figure 2: Why used A and B for both the panels and the curves. Consider changing to lower case letters to identify curves or even C1, C2 etc.

Experimental design

Generally, the experimental design and use of methods is sound. However, in the introduction and conclusions the study makes several untrue claims about being the first to quantify variation in river dolphin skull shape (e.g. Line 7, Line 62, Line 313).

Several studies have quantified variation in odontocete skull shape, including river dolphins. For instance, Barroso, Cranford, and Berta (2012) identified a river dolphin morphotype in mandible shape. Also, McCurry et al. (2017a) identified a river dolphin morphotype, tested for convergence and discussed the cause of this convergence. Neither of these were cited in your introduction, but appear later in your discussion.

There are several differences in your approach and sampling that allow your study to fill a knowledge gap, but being the “first quantitative investigation of the morphological variation present in the skulls of river dolphins” is not one of them. Please rewrite your introduction and conclusions as to better identify a knowledge gap and alter your conclusions in relation to this.

Validity of the findings

Overall, I think that the conclusions are scientifically sound and supported with enough evidence.

As stated above, and in the line edits, I think that the rational, benefit to the literature and conclusions need to be re-written.

As stated in the line edits I think there needs to be more clarification of why the species sampled were sampled.

Reviewer 2 ·

Basic reporting

The paper is generally clear and easy to follow. There is some detail lacking at times, but generally I see no issues with language or clarity of the text.

Figures 3 & 4 are hard to read. The dots are quite small, and don’t contrast from the other elements well enough. Correcting this would make the figures much clearer.

In Figure 2, I think it would be more helpful to show the placement of the semi-landmarks, as opposed to the outline only

Experimental design

The introduction is centred on the principle that morphological convergence in river dolphins as never been quantitatively assessed, however this has been done by other studies to some extent (Werth 2006; Barroso et al. 2012; Kelley and Motani 2015; McCurry et al. 2017a, b). In particular, the recent study by (McCurry et al. 2017a) was centred not only on the convergence between river dolphins, but also with crocodiles sharing a similar ecology. I suggest a better way to phrase this would be to state that although this has been done, a detailed analyses of the cranial features that converge (or not) in river dolphins is still lacking. This would also be more in line with the guidelines of the journal, of emphasising a gap in current knowledge rather than the novelty of a study per se.

Lines 98-99: there is usually more variation in the dorsal side of odontocete skulls than in the ventral, so the authors need to justify why this analyses focus only on the ventral side. I think I know why, but best to be explicit and clear so readers don’t have to guess.

References

Barroso C, Cranford TW, Berta A (2012) Shape analysis of odontocete mandibles: Functional and evolutionary implications. J Morphol 273:1021–1030. doi: 10.1002/jmor.20040

Kelley NP, Motani R (2015) Trophic convergence drives morphological convergence in marine tetrapods. Biol Lett 11:20140709. doi: 10.1098/rsbl.2014.0709

McCurry MR, Evans AR, Fitzgerald EMG, et al (2017a) The remarkable convergence of skull shape in crocodilians and toothed whales. Proc R Soc B Biol Sci 284:20162348. doi: 10.1098/rspb.2016.2348

McCurry MR, Fitzgerald EMG, Evans AR, et al (2017b) Skull shape reflects prey size niche in toothed whales. Biol J Linn Soc 121:936–946. doi: 10.1093/biolinnean/blx032

Werth AJ (2006) Mandibular and Dental Variation and the Evolution of Suction Feeding in Odontoceti. J Mammal 87:579–588. doi: 10.1644/05-mamm-a-279r1.1

Validity of the findings

I don’t think these results show independent convergence between all individual river dolphin lineages as implied, but rather convergence between Platanista and all other river dolphins. From the phylogenetic vs phenetic tree comparisons, only Platanista shifts its position relative to other river dolphins. These also group together in the molecular tree, so their placement is largely consistent in the phenetic tree.

The section where the possibility of plesiomorphy is discarded is not well justified. It’s not that the authors are not right in dismissing it necessarily, just that they don’t provide a convincing argument for it. I suggest the authors present more details to explain their position.


Lines 62-63: as per guidelines of PeerJ, statements like this highlighting the novelty of a study should be avoided. Also, please note my comments above as to why I feel this is not strictly true.

Additional comments

Overall, I think the manuscript is well written, and focus on an interesting scientific question regarding morphological evolution in cetaceans. The data analysis appears to be done competently, although I would like a bit more consideration of the phylogenetic uncertainty in molecular studies of river dolphins.

However, I think the narrative is slightly inconsistent as the paper states in the introduction this has never been done, and then cite in the discussion earlier papers that already suggested it. I think the paper would be much improved if the authors focused on the details of which aspects of the morphology change more, and how these fit with possible mechanisms driving the convergence. For this, I think it would be useful if the author could analyse the dorsal side of the skulls, where most of the variation is likely to be found.

Therefore I’m recommending the paper be rejected in its current form, but encourage a resubmission as the authors have the data to allow a more interesting analyses to be presented. In addition, please find below some minor comments on formatting and wording:

Lines 118-119: (McCurry et al. 2017a, b) also carried out GMM analyses on mandibles. This must be acknowledged

Lines 57-58: The authors probably mean cervical vertebrae, not caudal. Also, references to Figure 1B are not entirely appropriate, as Figure 1B does not show the vertebrae.

Lines 75-77: the phrasing here is unclear. It suggests that all species were photographed, but this is clearly not the case from the list in the supplementary material. If the authors meant that the specimens analysed cover all the odontocete families, then they need to specify that the family Kogiidae is also not represented.

Lines 81-83: Please specify how individual specimens were determined as being full grown adults.

References

McCurry MR, Evans AR, Fitzgerald EMG, et al (2017a) The remarkable convergence of skull shape in crocodilians and toothed whales. Proc R Soc B Biol Sci 284:20162348. doi: 10.1098/rspb.2016.2348

McCurry MR, Fitzgerald EMG, Evans AR, et al (2017b) Skull shape reflects prey size niche in toothed whales. Biol J Linn Soc 121:936–946. doi: 10.1093/biolinnean/blx032

---

## Round 0.2 · Minor Revisions

Thank you for your revised manuscript and clear responses to the reviewers' comments. For the most part I am satisfied with the revisions that you have made and am happy that they address all the reviewers' concerns. There are a few outstanding minor revisions that I think need to be made:

1. In response to a comment from reviewer 1, you amended a sentence at the beginning of the discussion from 'experienced evolution in the same direction' to 'evolved similar skull shapes'. Can you make the same revision in the abstract as well please? (Lines 13-14).

2. Line 94: 'Inidae' should be 'Iniidae'. Also, here you include Pontoporia in Iniidae, but in Figure 1A, Pontoporia is given its own family.

3. Lines 95-96: 'Plantanista forms a sister group to the remaining Odontoceti'. Remaining from what? This reads to me as if Plantanista is the sister group to all odontocetes except Iniidae and Lipotes. Even if it means sister group to all other odontocetes, it's not true because sperm whales are more basal.

4. Lines 97-99: 'Monodontidae (belugas and narwhals),Phocoenidae (porpoises) and Delphinidae (dolphins) are placed as sister groups to this clade and are therefore all phylogenetically equally related to the river dolphin genera.' This sentence confused reviewer 1 and it still confuses me. Firstly, it's not clear what 'this clade is'. Second, it sounds like you're saying Monodontidae et al are equally related to each of the four river dolphin genera, which is not true - they are more closely related to Inia, Pontoporia and Lipotes than they are to Plantanista. Are you trying to say that Monodontidae has the same relationships to the river dolphin genera as does Phocoenidae and Delphinidae? If so, it's not really coming across like that. To be honest, I think you can just discard the phrase about phylogenetically equal relatedness, and rewrite this section much more simply - Monodontidae, Phocoenidae and Delphinidae together form the sister-group to Iniidae plus Lipotes.

5. Line 99: 'We therefore sampled species (n = 24, Table S1, Figure S1) across all groups except sperm whales (Physeteridae)'. Remove 'therefore' as I don't think your sampling strategy is a direct consequence of the phylogeny here. Also, I presume you are including Kogia within Physeteridae? I realise there is some differences of opinion about this, so perhaps you can make it clear in the text.

Otherwise, I am happy with the manuscript. I don't think these minor revisions will take long to address, so I look forward to seeing a revised manuscript in the near future.

---

## Round 0.3 · accepted · Accept

Thanks for making those last few revisions so promptly. I am how happy to accept the manuscript for publication.